# Risk Evaluation for a Manufacturing Process Based on a Directed Weighted Network

**DOI:** 10.3390/e22060699

**Published:** 2020-06-23

**Authors:** Lixiang Wang, Wei Dai, Dongmei Sun, Yu Zhao

**Affiliations:** 1School of Reliability and Systems Engineering, Beihang University, Beijing 100191, China; wlx29@buaa.edu.cn (L.W.); sdmei208@163.com (D.S.); zhaoyu@buaa.edu.cn (Y.Z.); 2No. 208 Research Institute of China Ordnance Industries, Beijing 100191, China

**Keywords:** directed weighted network, PMIME, information entropy, risk evaluation

## Abstract

The quality of a manufacturing process can be represented by the complex coupling relationship between quality characteristics, which is defined by the directed weighted network to evaluate the risk of the manufacturing process. A multistage manufacturing process model is established to extract the quality information, and the quality characteristics of each process are mapped to nodes of the network. The mixed embedded partial conditional mutual information (PMIME) is used to analyze the causal effect between quality characteristics, wherein the causal relationships are mapped as the directed edges, while the magnitudes of the causal effects are defined as the weight of edges. The node centrality is measured based on information entropy theory, and the influence of a node is divided into two parts, which are local and indirect effects. Moreover, the entropy value of the directed weighted network is determined according to the weighted average of the centrality of the nodes, and this value is defined as the risk of the manufacturing process. Finally, the method is verified through a public dataset.

## 1. Introduction

In a product manufacturing process, due to the random fluctuations of process factors, deviations in quality characteristics occur. Moreover, random fluctuations are inherent in all stages of the manufacturing process and are difficult to avoid. Therefore, quality control of the manufacturing process has always been an active and important topic [1]. During the manufacturing process, there is a complex coupling relationship between quality characteristics. In general, machining errors for one key characteristic may cause some errors for other characteristics coupled with the key characteristic. Therefore, identifying key quality characteristics is particularly important for quality control in the manufacturing process. Amiri et al. [2] believed that the quality of a manufacturing process is represented by two or more correlated quality characteristics. Obviously, analyzing the relationship between quality characteristics in isolation may lead to erroneous results. 

Many scholars have studied the relations between quality characteristics for product design. Quality Function Deployment (QFD) is a common theory and method for product design management driven by customer requirements [3]. The QFD is a process and set of tools used to effectively define customer requirements and convert them into quality characteristics. However, the relations in the design phase are static, this is because the analysis of the quality characteristics of the design process is performed before the product is manufactured. During the manufacturing process, quality characteristics are constantly changing due to changes in many process factors. Therefore, it is not enough to analyze the relationship of quality characteristics from a static state. At the same time, the constant changes in quality characteristics have also brought uncertainty to the risk evaluation of the manufacturing process. Some quality control methods are presented for the multistage manufacturing process, such as the quality state space theory [4] and stream of variation theory [5]. These methods analyze the change of quality characteristics with the process from the perspective of a time series. It is known that the accumulation of deviations in quality characteristics leads to defects, which are then exposed in the form of early failures during use. Moreover, the relation between quality characteristics is more precisely a causal relationship, because the accumulation of quality characteristic deviations in the previous stage will propagate during the manufacturing process, thereby affecting the quality characteristics in the following stage.

In this paper, the relation of quality characteristics is considered as a directed weighted complex network. The quality characteristics of each process are mapped to nodes of the network. And the partial conditional mutual information (PMIME) is used to analyze the causal effect between quality characteristics of the manufacturing process. Further, the causal relationship between quality characteristics are mapped as a directed edge, while the magnitudes of the causal effects are defined as the weight of edges, thereby a directed weighted network is established. Moreover, the centrality of a node is measured based on information entropy theory. The influence of a node is divided into two parts, which are local and indirect effects. The larger the value of entropy of a node, the greater its influence. Moreover, the entropy *H*(*I*) of a single discrete random variable *I* is a measure of its average uncertainty. For the set of quality characteristics, the entropy of each quality characteristic represents the uncertainty of whether it can complete the requirements of the manufacturing process. That is, the centrality of each node indicates the uncertainty of whether the quality characteristics can meet the requirements of the manufacturing process in a certain time series. In addition, the quality of the manufacturing process actually refers to the degree to which a set of quality characteristics meets its production needs. Therefore, the risk of the manufacturing process is defined as the quality loss caused by the quality characteristics not meeting the production requirements. In this paper, the risk is evaluated by quantifying the uncertainty of the manufacturing process. When the uncertainty of the manufacturing process is greater, the more defects in the manufacturing process and the greater the quality loss. Furthermore, the quality loss is invisible, which means that the reliability of the products produced by the manufacturing process is relatively low. Moreover, the quality loss will spread over the connection between quality characteristics, so the connection between quality characteristics is actually the risk propagation path of the manufacturing process. Therefore, the entropy value of the directed weighted network is determined according to the weighted average of the centrality of the nodes, and the value is defined as the risk of the manufacturing process. Finally, the method is verified through a public dataset.

The outline of this paper is organized as follows. In Section 2, a literature review related to complex network theory is applied to solve and describe complicated manufacturing problems and causality analyses are presented. Section 3 includes details of the proposed approach. Section 4 contains details about a real case to illustrate and verify the proposed method. Finally, the conclusions drawn and discussion are in Section 5.

## 2. Literature Review 

Due to the random fluctuation of process factors, which makes the quality characteristics uncertain, there is a complex causal relationship between quality characteristics in a time series. Therefore, it is essential to establish a model to describe the behavior of the manufacturing process. Some mathematical models that are proposed by researchers that describe the quality information flow in a multistage manufacturing process, the assembly process [6,7] and machining process [8] are included. Hu [9] presented the stream of variation theory for automotive body assembly. Then the variation flow theory was extended to the machining process [10,11,12]. However, in the actual manufacturing process, quality characteristics affect each other. Therefore, analysis of the error transfer and accumulation of quality characteristics in a multistage manufacturing process may lead to erroneous results. Jiao and Djurdjanovic [13] presented the compensability of error in product quality to eliminate quality errors accumulated in a multistage manufacturing process. These methods provide some opportunities for the development of modeling and quality control in a multistage manufacturing process.

Many researchers have applied complex network theory to describe and solve complicated manufacturing problems. Based on the complexity of multistage processes, some network models have been established to improve product quality. Lin and Chang [14] focused on performance evaluation of a manufacturing system with multiple production lines based on the network-analysis perspective. Wang et al. [15] proposed an extended machining error propagation network model to quantitatively analyze the complex coupling relationship in the small-batch, multistage machining process of aircraft landing gear parts. Liu et al. [16] used a machining error propagation network of multistage machining processes to describe complicated interactions among different stages. Cheng and Chu [17] proposed a network-based assessment approach for change impacts on complex products, and three changeability indices were presented. Qin et al. [18] utilized a weighted network of multistage machining processes to quantitatively analyze variation propagation. Moreover, the variation propagation stability was analyzed by a virus-spreading model. Kim et al. [19] proposed a product network analysis, which explored a network-leveled relation among all products. Wang et al. [20] proposed a novel approach to support failure mode, effects, and criticality analysis for multistage processes based on complex networks. Di Bona et al. [21] proposed a total efficient risk priority number method that integrated the failure mode, effects, and criticality analysis with other important factors in risk assessment. The above complex network-based modeling analyzed the quality evaluation and control of multistage processes from different levels. 

Nevertheless, there is a lack of analysis using quality characteristics to describe the quality of multistage processes. Actually, the accumulation of deviations in quality characteristics leads to the occurrence of defects, which are exposed in the form of early failures during the use phase. Moreover, the quality of a product or process is characterized by monitoring correlated profile and multivariate quality characteristics. Therefore, it is important to analyze the behavior of multistage processes using quality characteristics. Du et al. [22] defined the key product characteristics and designed a model for a key characteristics management system. Köksoy [23] presented a method to optimize multiple quality characteristics based on the mean square error criterion. Ouyang et al. [24] used a QCAC-Entropy-TOPSIS approach to measure quality characteristics and rank improvement priorities for all substandard quality characteristics. Li et al. [25] proposed a key quality characteristics selection technique for imbalanced production data using a two-phase, bi-objective feature selection method. Diao et al. [26] analyzed the coupling relations among quality characteristics and proposed a weighted-coupled, network-based quality control method for improving key characteristics in the product manufacturing process. 

However, these methods ignore the causal relationship between quality characteristics and the direction of causal effects. This is because the accumulation of deviations in quality characteristics will propagate during the manufacturing process and affect the quality characteristics in the next stage. Generally, Granger causality [27] and transfer entropy [28,29] are two classic methods for causal analysis. However, these two classic causal analysis methods are only suitable for a bivariate time series. With the development of multivariate state space reconstruction, different embedding schemes [30,31,32,33] are used in Granger causality and transfer entropy. Multivariate time series embedding includes uniform and non-uniform embedding. Uniform embedding may have problems such as overfitting and false influence detection. A nonuniform embedding scheme solves the above problems perfectly [34]. Vlachos et al. [35] presented a causality measure of conditional mutual information from mixed embedding (MIME) for bivariate time series. Further, Kugiumtzis [36] extended the MIME method to a multivariate time series, which is named as partial MIME (PMIME).

## 3. Proposed Method

### 3.1. Preliminaries 

In a multistage manufacturing process, there is a complex causal relationship between quality characteristics. Moreover, due to the random fluctuation of process factors, each quality characteristic is uncertain in a time series. Therefore, if we want to study the effect of the causal relationship between quality characteristics on the quality of manufacturing process, we should first model the multistage manufacturing process and extract the corresponding quality characteristics. Suppose that *k* is the number of machining stages, and the stage set is written as $S={\{S}_{1}{,\;S}_{2}{,\;\ldots \;,\;S}_{k}\}$. Moreover, $\varepsilon ={\{\varepsilon }_{1}{,\;\varepsilon }_{2}{,\;\ldots \;,\;\varepsilon }_{k}\}$ and $\tau ={\{\tau }_{1}{,\;\tau }_{2}{,\;\ldots \;,\;\tau }_{k}\}$ are defined as the number of individual quality characteristics extracted for a single machining stage and the length of time for each machining stage. Hence, a model of a multistage manufacturing process is shown in Figure 1.

Suppose that *n* is the number of types of quality characteristics, and the characteristic set is written as $={\{C}_{1}{,\;C}_{2}{,\;\ldots \;,\;C}_{n}\}$. Therefore, quality characteristics are extracted from the multistage manufacturing process. This is shown in Figure 2.

Therefore, a directed weighted network is represented as following.
(1)G=(N, E, W)
where ${N=\{N}_{1}{,\;N}_{2}{,\;\ldots \;,\;N}_{n}\}$, and *N* denotes the set of nodes in a given directed weighted network, ${E=\{e}_{11}{,\;e}_{12}{,\;\ldots \;,\;e}_{ij}{,\;\ldots \}}_{n\times n}(1\leq \;i\;\leq \;n,\;1\leq \;j\;\leq \;n)$, and *E* represents the set of directed edges from one node to another, ${W=\{w}_{11}{,\;w}_{12}{,\;\ldots \;,\;w}_{ij}{,\;\ldots \}}_{n\times n}(1\leq \;i\;\leq \;n,\;1\leq \;j\;\leq \;n)$, and set *W* corresponds to the weighted values. In this paper, nodes are denoted as the quality characteristics of a multistage manufacturing process. Obviously, there is a one-to-one correspondence between the characteristic set *C* and the node set *N*. The edges set represents causality among quality characteristics i→ j, and set *W* of weighted values indicates the magnitude of the causal effect sent from quality characteristic i→ j. When $w_{ij}\;>\;0$, which means that node i has a causal effect on node j, and also means edge $e_{ij}$ exists. Otherwise $w_{ij}=0$, which means that node i has no causal effect on node j, and also means edge $e_{ij}$ does not exist. We assume a set of quality characteristics for a multistage manufacturing process, where $C={\{\;C}_{1}{,\;C}_{2}{,\;C}_{3}{,\;C}_{4}{,\;C}_{5}\;\}$. A simple example of a directed weighted network is shown in Figure 3.

### 3.2. PMIPE Method

In this paper, the PMIME method is used to estimate the direct and directional coupling in a multivariate time series. Let ${{\{x}_{t}{,\;y}_{t}{,\;z}_{1,t}{,\;\ldots \;,\;z}_{n-2,t}\}}_{t=1}^{\overset{k}{\underset{i=1}{\sum }}\tau_{i}}$ be a multivariate time series of *n* variables *X*, *Y*,${\;Z}_{1}$, $Z_{2},$ …, $Z_{n-2}$. What we intend to estimate is the casual effect of *X* on *Y* conditioning on ${Z=\{\;Z}_{1}$, $Z_{2},\;$…, $Z_{n-2}\}$. Hence, the driving subsystem is *X* and the response subsystem is *Y*, and *Z* is defined as the other subsystem. Obviously, subsystems *X*, *Y*, Z contain all quality characteristics of the multistage manufacturing process.

Moreover, the future of variable *X* is represented by a vector of *T* future values, that is $x_{t}^{T}{=[x}_{t+1}{,\;x}_{t+2}{,\;\ldots \;,\;x}_{t+T}],1\leq T\leq \overset{k}{\underset{i=1}{\sum }}\varepsilon_{i}$. In addition, the lags of *X*, *Y* and *Z* are sought in a range given by a maximum lag for each variable, e.g., $L_{x}$ for *X* and $L_{y}$ for *Y*. Generally, the maximum lag *L* of all variables is equal, that is $L_{x}{=L}_{y}{=L}_{z}.$
${\;V}_{t}$ is indicated as the set of all lagged variables at time *t*, the part of $x_{t}{,\;x}_{t-1}{,\;\ldots \;,\;x}_{{t-L}_{x}}$ of *X* and the same for subsystem *Y* and *Z* are contained in ${\;V}_{t}$. An iterative technique is performed to form the mixed embedding vector ${\;v}_{t}\;\in {\;V}_{t}$. The detailed method is shown in the following steps. 

Step 1: starting with an empty embedding vector $v_{t}^{0}=\emptyset \;$. 

Step 2: in the first iteration in order to find the most related to $y_{t}^{T}$ in ${\;V}_{t}$, the embedding vector is represented as $v_{t}^{1}$, which is written as follows:(2)$$v_{t}^{1}=\underset{v\;\in {\;V}_{t}}{argmax}{I(y}_{t}^{T};\;v)$$
where *I*(.) denotes mutual information (MI). And MI is estimated by the k-nearest neighbors (k-NNs) method. Then $w_{t}^{1}{=[v}_{t}^{1}]$ is obtained, simultaneously $v_{t}^{1}$ is removed from ${\;V}_{t}$.

Step 3: in the *m*-th (*m* ≥ 2) embedding cycle, the mixed embedding vector is augmented by the component $v_{t}^{m}$ of ${\;V}_{t}$, giving most information about $y_{t}^{T}$ additionally to the information already contained in $w_{t}^{m-1}{=[v}_{t}^{1}{,\;v}_{t}^{2}{,\;\ldots \;,\;v}_{t}^{m-1}]$. As for step 2, $v_{t}^{m}$ is denoted as follows:(3)$$v_{t}^{m}=\underset{v\;\in {\;V}_{t}}{argmax}{I(y}_{t}^{T}{;\;v|w}_{t}^{m-1})$$

For example, in the second iteration, $v_{t}^{2}$ is written as follows:(4)$$v_{t}^{2}=\underset{v\;\in {\;V}_{t}}{argmax}{I(y}_{t}^{T}{;\;v|w}_{t}^{1})$$
where the conditional mutual information (CMI) is calculated by the k-NNs method, and the mixed embedding vector is $w_{t}^{2}{=[v}_{t}^{1}{\;,\;v}_{t}^{2}]$. Iterating occurs according to Equation (3) until the additional information of $v_{t}^{m}$ selected at the embedding cycle *m* is not large enough. Moreover, the termination criterion is quantified as:(5)$${I(y}_{t}^{T}{;\;w}_{t}^{k-1}{)\;/\;I(y}_{t}^{T}{;\;w}_{t}^{k})\;>\;A$$
where the threshold *A* < 1 and the value of *A* is generally 0.95 or 0.97 in [35,36]. Further, the mixed embedding vector $w_{t\;}{=w}_{t}^{k-1}$ is obtained.

Step 4: calculating the causality effect of *X* on *Y* conditioned on the other variables in Z, the PMIME is described as
(6)$$R_{X\to Y|Z}=I\left(y_{t}^{T}{;\;w}_{t}^{x}{|w}_{t}^{y}{,\;w}_{t}^{z}\right){\;/\;I(y}_{t}^{T}{;\;w}_{t})$$
where ${\;w}_{t}^{x}$ is denoted as the part of *X* in ${\;w}_{t}$, and the same with $w_{t}^{y}\;$ and ${\;w}_{t}^{z}$.

### 3.3. Entropy-Based Centrality Measurement

Based on the PMIME method, the direction of the causal relation and the magnitude of the causal effect of each quality characteristic can be determined. Moreover, each quality characteristic is mapped to a node, the direction of the causal relationship between nodes is mapped to directed edges, and the magnitude of the causal effect between nodes is mapped to the weight of the edges. Hence, a directed weighted network is defined. Furthermore, Qiao et al. [37] proposed an entropy-based centrality measurement to identify the vital node. The total power of a node is divided into two parts, including its local power and its indirect power. The detailed method is described below as follows:

Step 1: a complete directed weighted network is deconstructed into several subnets centered on certain nodes.

Step 2: calculating structural entropy (SE), the SE takes advantage of topographic properties of the sub-graph, evaluating the strength of a given node in specific subnet. Above all, the subnet degree centrality of node *i* and its neighbor *j*, which is indicated as ${SDC}_{i}$. This is written as
(7)$${SDC}_{i}{=DC}_{i}^{in}{+DC}_{i}^{out}$$
where ${\;DC}_{i}^{in}$ denotes the in-degree centrality of node *i* (the number of nodes pointing to node *i*) and ${\;DC}_{i}^{out}$ represents the out-degree centrality of node *i* (the number of directed edges from node *i* to another node). Moreover, the *SE* for node *i* in subnet $G_{i}$ is indicated as follows:(8)$${SE}_{i}=-\sum_{i=1}^{M+1}\frac{{SDC}_{i}}{\sum_{i=1}^{M+1}{SDC}_{i}}log\frac{{SDC}_{i}}{\sum_{i=1}^{M+1}{SDC}_{i}}$$
where *M* refers to the number of nodes directly connected to node *i* in subnet $G_{i}$.

Step 3: calculating frequency entropy (FE), the FE takes advantage of the weighted edges that reflects the interaction frequency between two nodes. Further, the FE for node *i* in subnet $G_{i}$ is stated as follows:(9)$${FE}_{i}=-\sum_{j=1}^{H}\frac{W_{ij}}{\sum_{k=1}^{H}W_{ik}}log\frac{W_{ij}}{\sum_{k=1}^{H}W_{ik}}$$
where $W_{ij}$ indicates the weight of a directed edge in the given direction and *H* is the number of node *i* points to other nodes.

Step 4: Combining Equations (8) and (9), the local power of node *i* is denoted, which is named as ${LE}_{i}$. Moreover, the ${LE}_{i}$ is stated as follows:(10)$${LE}_{i}{=\theta }_{1}{SE}_{i}{+\theta }_{2}{FE}_{i}$$
where $\theta_{1}$ and ${\;\theta }_{2}$ are the weight coefficients respectively, and $\theta_{1}{+\theta }_{2\;}=1.$

Step 5: Calculating the indirect power of node *i* on its second-order neighbor node *k*, which is denoted as ${IE}_{ik}$. This is written as follows:(11)$${IE}_{ik}=-\sum_{k=1}^{N_{ik}}\frac{{LE}_{i}{\times LE}_{s}}{N_{ik}}$$
where $N_{ik}$ is the number of first-order neighbor nodes between node *i* and *k*. And ${LE}_{s}$ represents the local power of node *s*, and node *s* connects nodes *i* and *k*. We take a two-path subnet with a quadrilateral structure as an example, which is shown in Figure 4.

As described above, node *s* represents node *j* and *l* in Figure 4. Moreover, the indirect power of node *i* on its second-order neighbor node *k* is denoted as follows:(12)$${IE}_{ik}=\;\frac{{LE}_{i}\times {LE}_{j}+{LE}_{i}\times {LE}_{l}}{2}$$

Hence, the total indirect power of node *i* on its second-order neighbor nodes is denoted as ${IE}_{i}$, which is stated as follows:(13)$${IE}_{i}=\;\frac{\sum_{k=1}^{H_{i}}{IE}_{ik}}{H_{i}}$$
where $H_{i}$ is the total number of second-order neighbor nodes of node *i*.

Step 6: in line with Equations (10) and (13), the total power of node *i* is represented by $E_{i}$, which is named as follows:(14)$$E_{i}{=\mu }_{1}{LE}_{i}{+\mu }_{2}{IE}_{i}$$
where $\mu_{1}$ and ${\;\mu }_{2}$ are the weight coefficients respectively, and $\mu_{1}{+\mu }_{2}=1.$

### 3.4. Risk Evaluation

Burduk and Chlebus [38] thought of risk as the danger of failing to achieve the goals specified in the decision. In a multistage manufacturing process, the quality can be denoted as the sum of the characteristics of the process capability to meet explicit and implicit needs. Moreover, entropy is a measure of the uncertainty of the state of quality characteristics in the manufacturing process, that is, a measure of quality loss. Hence, risk is defined as the quality entropy of a multistage manufacturing process. Moreover, since the multistage manufacturing process can be represented by a directed weighted network, the risk of the multistage manufacturing process is defined as the weighted average of the centrality of the nodes. Further, the node weight is calculated as follows:(15)$$\lambda_{i}=\frac{{SDC}_{i}}{\sum_{i=1}^{N}{SDC}_{i}}$$
where, ${SDC}_{i}$ represents the degree of node *i* and *N* is the number of nodes. In addition, $\overset{N}{\underset{i=1}{\sum }}\lambda_{i\;}=1$. Thus, the risk of a multistage manufacturing process is written as follows:(16)$$R=\sum_{i=1}^{N}\lambda_{i}E_{i}$$

## 4. Case Study

The data of the case come from the SECOM dataset of the UCI Machine Learning Repository [39], which is about a semi-conductor manufacturing process and has 1567 samples, each sample with 591 quality characteristics. The first 59 quality characteristics of the data set were extracted as an example to illustrate the algorithm in this paper. In other words, $\overset{k}{\underset{i=1}{\sum }}\tau_{i}=1567$ and n=59. Moreover, causality between 59 quality characteristics was determined based on PMIME, and we used A=0.95, L=5  and T=3. Further, a directed weighted network was setup, which is shown in Figure 5.

Where the yellow line indicates that the weight of the edge is greater than 0.5, and the blue line indicates that the weight of the edge is less than or equal to 0.5. Moreover, nodes 18, 50, and 53 are isolated and not connected to other nodes. In addition, the degree distribution of nodes is shown in Figure 6.

According to Figure 6, it can be concluded that the degree distribution of nodes does not have much regularity, which is neither like Poisson distribution nor power-law distribution. More samples may be needed to further observe the statistical characteristics of the degree distribution. Moreover, only a few nodes have a higher degree. We took the subnet of node 1 as an example to explain the calculating process of the proposed algorithm, and the results of ${SDC}_{i}$ as shown in Table 1.

Based on Equation (8), 10 is set as the base of the logarithmic function, then the structural entropy of node 1 is described as follows:(17)$${SE}_{1}=-\sum_{i=1}^{10}\frac{{SDC}_{i}}{\sum_{i=1}^{10}{SDC}_{i}}log\frac{{SDC}_{i}}{\sum_{i=1}^{10}{SDC}_{i}}=0.9774$$

Moreover, the frequency entropy is calculated by Equation (9), which is written as follows:(18)$${FE}_{1}=-\sum_{j=1}^{6}\frac{W_{ij}}{\sum_{k=1}^{6}W_{ik}}log\frac{W_{ij}}{\sum_{k=1}^{6}W_{ik}}=0.7108$$

In Equation (10), the $\theta_{1}$ and $\theta_{2}$ are set as 0.4 and 0.6, respectively and the local influence of node 1 is defined as follows:(19)$${LE}_{1}={0.4\;SE}_{1}+{0.6\;FE}_{1\;}=0.8174$$

Following Equations (11) and (13), the indirect influence of node 1 is stated as follows:(20)$${IE}_{1}=\;\frac{\sum_{k=1}^{30}{IE}_{ik}}{30}=0.9242$$

Further, the $\mu_{1}$ and $\mu_{2}$ are denoted as 0.6 and 0.4 particularly. Then the total influence of node 1 is expressed as follows:(21)$$E_{1}={0.6\;LE}_{1}+{0.4IE}_{1}=0.8601$$

Hence, the total power of each node and the corresponding ranking results are shown in Table 2.

Further, the comparison of the local power, indirect power, and total power of each node is shown in Figure 7.

Combining Table 2 and Figure 7, in the whole network, nodes 3, 4, 29, 34, 41, and 52 are obviously more important than the other nodes. Moreover, the values of entropy-based centrality of these six nodes are all bigger than 1, while the total power of the 7th ranked node is 0.8940, which is a big gap with the top six nodes. Hence, nodes 3, 4, 29, 34, 41, and 52 are considered as vital nodes, for their changes have a greater influence on the nature of the directed weighted network, and they affect more nodes, too. Moreover, the value of entropy-based centrality of 32 nodes is less than 0.5, which is more than half of the data set. And only nine nodes are greater than 0.8, which includes nodes 1, 3, 4, 23, 29, 34, 37, 41 and 52. This is consistent with the information given in Figure 6. That is, only a few nodes have a higher degree and can affect more nodes. Therefore, the risk of the manufacturing process can be reduced by improving the quality of key nodes. Moreover, the risk of the manufacturing process is defined by the weighted average of the centrality of each node. The weight of each node is obtained through the degree of the node. The greater the degree of the node, the higher the weight, and the greater the probability that the node poses a risk to the whole manufacturing process. Further, based on Equation (15), the degree of each node and their weights are shown in Table 3.

Furthermore, following Equation (16), the risk of the whole network is defined as follows:(22)$$R=\sum_{i=1}^{59}\lambda_{i}E_{i}=0.6390$$

In addition, as shown in Table 2, the biggest value of entropy-based centrality of nodes is 1.2108. Hence, the risk scope of the whole manufacturing process is from 0 to 1.2108. Supposing the range is divided into three sets from small to large, which is [0, 0.4036), [0.4036, 0.8072) and [0.8072, 1.2108]. The corresponding risks are low, medium, and high, therefore the risk of manufacturing process is medium in this case. Moreover, nodes 3, 4, 29, 34, 41, and 52 account for 25.77% of the weight set. Obviously, controlling these six nodes is critical to improving the quality of the manufacturing process. 

## 5. Conclusions and Discussion 

A key quality characteristic affects many other quality characteristics in a multistage manufacturing process, and the fluctuation of key quality characteristics in the manufacturing process makes the quality characteristics affected by it deviate. Therefore, analyzing the individual quality characteristics in isolation may cause a large deviation in the risk of the manufacturing process. Moreover, the quality characteristics are constantly changing over time. In this paper, the set of quality characteristics is divided into three subsystems: the driving subsystem, response subsystem, and the other subsystem. And the PMIME is used to mine the causality between quality characteristics in the time series. Further, based on complex networks theory, the causal relationship between quality characteristics is mapped to a directed edge, while individual quality characteristics are mapped to nodes, and the magnitudes of the causal effects are defined as the weight of edges. Then a directed weight is established, and the power of a node is divided into two parts, which are local and indirect effects. An entropy-centrality approach is applied to rank influential nodes. This method innovatively solves the possible problem of determining the edges of the complex network through static qualitative analysis because the status of nodes and the status between nodes are updated at any time. In addition, the quality of the manufacturing process is represented by two or more correlated quality characteristics. Hence, a novel index for evaluating risk based on the entropy-centrality of nodes of the complex network has been proposed. This indicator reflects the risk of the manufacturing process, and its size is constantly updated with the change of the state of the nodes and the magnitude and direction of the edge. Therefore, the risk of the manufacturing process can be better controlled according to the change of indicator.

According to Equation (22), the risk of the manufacturing process is 0.6390. First of all, we should determine the acceptable threshold of risk according to the actual situation. If the risk is acceptable, no risk control measures are required for the manufacturing process. Otherwise, we should take risk control measures for the manufacturing process. Second, nodes 3, 4, 29, 34, 41, and 52 are identified as key nodes, and controlling the risks of these six nodes can effectively reduce the risk of the entire manufacturing process. According to the subnets of these six nodes, other quality characteristics that are causally related to these six quality characteristics can be quickly determined. We can change the processing technology, etc., so that the causal chain of quality characteristics is interrupted, that is, the propagation path of the risk is cut off. For example, node 1 affects node 21, and node 21 has an effect on other nodes. As long as the impact on node 1 or node 21 is cut off, the risk propagation path from node 1 to node 21 is interrupted, and the risk of node 1 is reduced. The example can be used to verify the effectiveness of our proposed risk control strategy. Hence, the structural entropy of node 1 is described as follows:(23)$${SE}_{1}=-\sum_{i=1}^{9}\frac{{SDC}_{i}}{\sum_{i=1}^{9}{SDC}_{i}}log\frac{{SDC}_{i}}{\sum_{i=1}^{9}{SDC}_{i}}=0.9271$$

And the frequency entropy of node 1 is calculated as follows:(24)$${FE}_{1}=-\sum_{j=1}^{5}\frac{W_{ij}}{\sum_{k=1}^{5}W_{ik}}log\frac{W_{ij}}{\sum_{k=1}^{5}W_{ik}}=0.6235$$

Hence, the local influence of node 1 is 0.7449. Moreover, the indirect influence of node 1 is stated as follows:(25)$${IE}_{1}=\;\frac{\sum_{k=1}^{29}{IE}_{ik}}{29}=0.8806$$

Further, the total influence of node 1 is 0.7992 and the structural entropy of node 21 is changed as 0.8814. Then the local influence and the indirect influence of node 21 is 0.5290 and 0.5513, respectively. Further, the total influence of node 21 is 0.5379. In addition, the weights of all nodes will also change, which is shown in Table 4.

Finally, the risk of the manufacturing process is updated to 0.6375, which is lower than before. Since there are many edges between nodes in this paper, although the risk propagation paths of nodes 1 and 21 are cut off, the role of risk control for the entire manufacturing process is relatively small. At the same time, it also proves that cutting off the risk propagation path of node 1 and node 21 has a practical effect on the risk control of the manufacturing process. Moreover, similar risk control measures are continuously taken for other quality characteristic causal chains until the risk of the manufacturing process reaches an acceptable level.

In this study, PMIME is used to detect the direct causality in quality characteristics, however, the curse of dimensionality, resulting in inaccurate estimates of mutual information as the embedding space increases, is unavoidable. Thus, for future work, we expect to carry out further work on improving the accuracy of the algorithm under the premise of dimensionality reduction. In addition, the algorithm in this paper is suitable for a large sample size, but may not be suitable for a small sample size. Therefore, exploring the establishment of a complex network in the case of small samples and ranking of the importance of nodes is also a question worthy of further investigation. Moreover, identifying how process factors affect the quality of the manufacturing process is also a topic worth studying in future work. That is, what is the mechanism of the quality characteristic deviation caused by the process factors, which will have a great impact on the quality control? Furthermore, the PMIME method in this paper can be used for causality analysis between process factors and quality characteristics, which will make the research in this paper more significant.

## Figures and Tables

**Figure 1 entropy-22-00699-f001:**
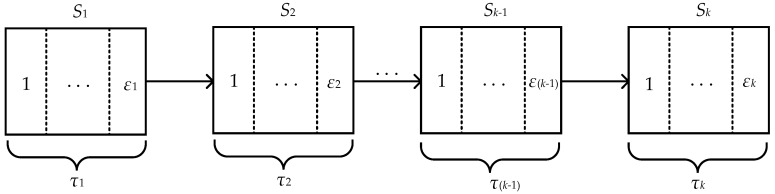
A model of a multistage manufacturing process.

**Figure 2 entropy-22-00699-f002:**
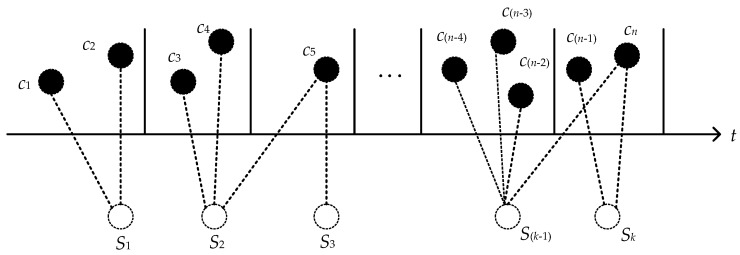
Extracting quality characteristics from a multistage manufacturing process.

**Figure 3 entropy-22-00699-f003:**
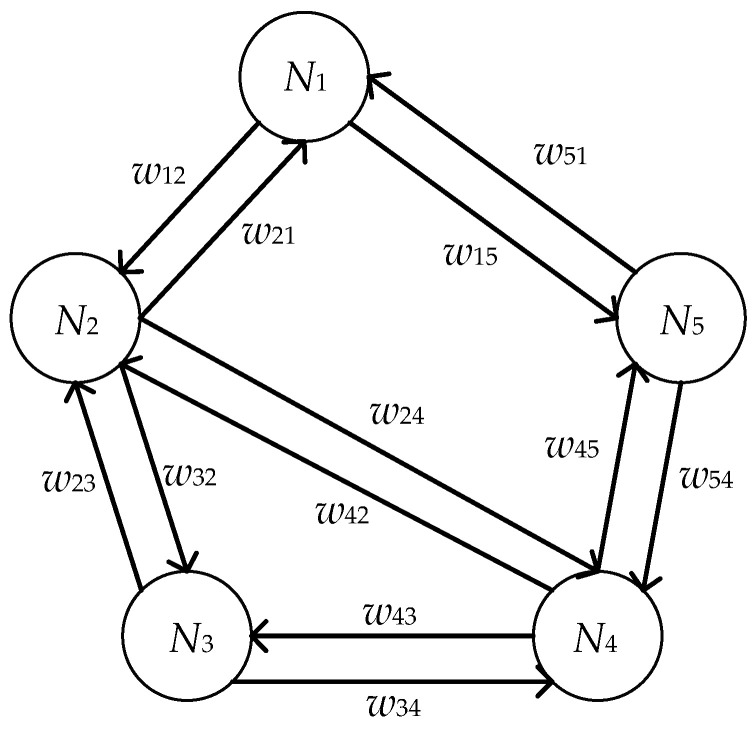
A simple example of a directed weighted network.

**Figure 4 entropy-22-00699-f004:**
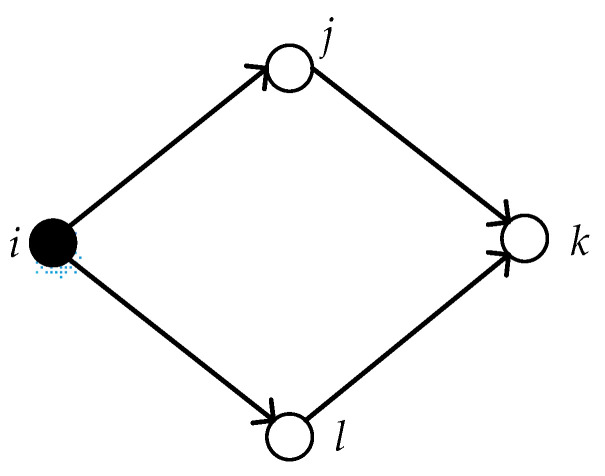
A two-path subnet.

**Figure 5 entropy-22-00699-f005:**
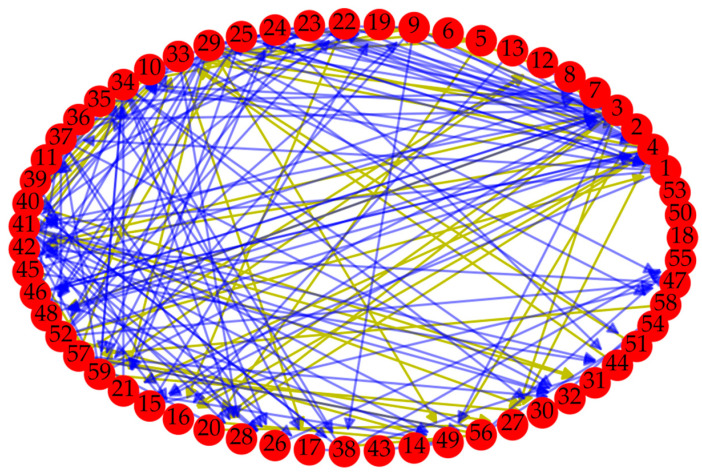
The directed weighted network of case.

**Figure 6 entropy-22-00699-f006:**
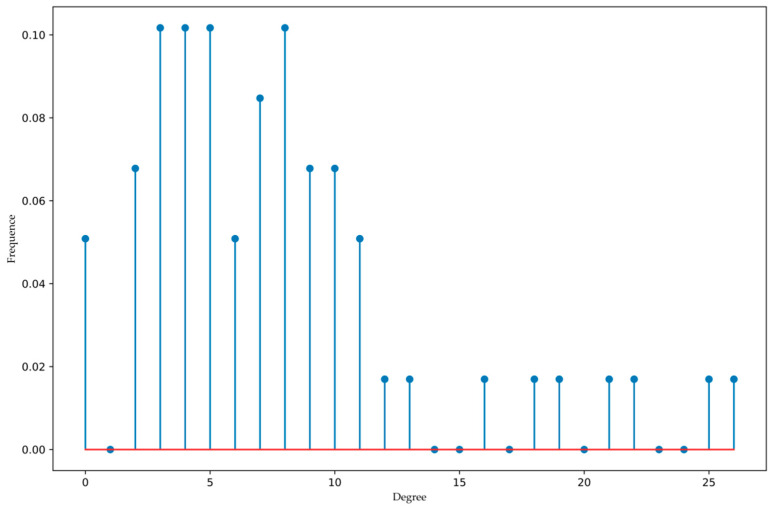
The degree distribution of nodes.

**Figure 7 entropy-22-00699-f007:**
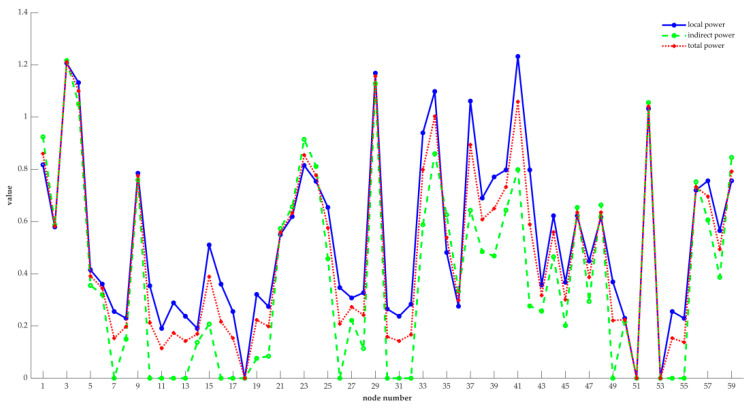
The comparison of the local power, indirect power, and total power of each node.

**Table 1 entropy-22-00699-t001:** The results of ${SDC}_{i}\;$ of nodes in subnet $G_{1}$.

Node	${DC}_{i}^{in}$	${DC}_{i}^{out}$	${SDC}_{i}$
1	4	6	10
4	2	3	5
21	5	1	6
23	4	2	6
25	4	0	4
29	3	0	3
34	4	1	5
37	3	4	7
41	1	7	8
52	1	7	8

**Table 2 entropy-22-00699-t002:** The total power of each node and the corresponding ranking results.

Node	${LE}_{i}$	${IE}_{i}$	$E_{i}$	No.	Node	${LE}_{i}$	${IE}_{i}$	$E_{i}$	No.
1	0.8174	0.9242	0.8601	8	31	0.2373	0	0.1424	51
2	0.5791	0.5867	0.5821	23	32	0.2833	0	0.1670	48
3	1.2070	1.2165	1.2108	1	33	0.9396	0.5878	0.7989	10
4	1.1320	1.0504	1.0994	3	34	1.0982	0.8598	1.0028	6
5	0.4150	0.3554	0.3912	29	35	0.4815	0.6256	0.5392	27
6	0.3607	0.3206	0.3447	32	36	0.2756	0.3326	0.2984	35
7	0.2555	0	0.1533	50	37	1.0614	0.6429	0.8940	7
8	0.2295	0.1497	0.1976	45	38	0.6898	0.4848	0.6078	21
9	0.7851	0.7593	0.7748	13	39	0.7708	0.4683	0.6498	17
10	0.3538	0	0.2123	42	40	0.7977	0.6437	0.7325	15
11	0.1908	0	0.1145	53	41	1.2324	0.7986	1.0589	4
12	0.2889	0	0.1734	46	42	0.7977	0.2768	0.5893	22
13	0.2373	0	0.1424	51	43	0.3571	0.2571	0.3171	33
14	0.1908	0.1374	0.1694	47	44	0.6222	0.4650	0.5593	25
15	0.5103	0.2072	0.3891	30	45	0.3673	0.2016	0.3010	34
16	0.3601	0	0.2161	41	46	0.6222	0.6539	0.6349	19
17	0.2555	0	0.1533	50	47	0.4486	0.2941	0.3868	31
18	0	0	0	54	48	0.6166	0.6633	0.6353	18
19	0.3210	0.0762	0.2231	39	49	0.3689	0	0.2213	40
20	0.2746	0.0844	0.1985	44	50	0	0	0	54
21	0.5498	0.5730	0.5591	26	51	0.2295	0.2156	0.2240	38
22	0.6191	0.6564	0.6340	20	52	1.0325	1.0557	1.0412	5
23	0.8151	0.9143	0.8548	9	53	0	0	0	54
24	0.7538	0.8111	0.7767	12	54	0.2555	0	0.1533	50
25	0.6542	0.4576	0.5756	24	55	0.2295	0	0.1377	52
26	0.3470	0	0.2082	43	56	0.7201	0.7518	0.7328	14
27	0.3074	0.2214	0.2730	36	57	0.7560	0.6060	0.6960	16
28	0.3274	0.1136	0.2419	37	58	0.5651	0.3868	0.4938	28
29	1.1683	1.1283	1.1563	2	59	0.7560	0.8454	0.7918	11
30	0.2649	0	0.1589	49					

**Table 3 entropy-22-00699-t003:** The degree of each node and their weights.

Node	${SDC}_{i}$	$\lambda_{i}$	Node	${SDC}_{i}$	$\lambda_{i}$	Node	${SDC}_{i}$	$\lambda_{i}$
1	10	0.0206	21	8	0.0165	41	25	0.0515
2	7	0.0144	22	7	0.0144	42	11	0.0227
3	26	0.0536	23	12	0.0247	43	2	0.0041
4	21	0.0433	24	8	0.0165	44	7	0.0144
5	3	0.0062	25	9	0.0186	45	6	0.0124
6	2	0.0041	26	7	0.0144	46	10	0.0206
7	4	0.0082	27	5	0.0103	47	5	0.0103
8	3	0.0062	28	6	0.0124	48	10	0.0206
9	9	0.0186	29	18	0.0371	49	8	0.0165
10	7	0.0144	30	42	0.0866	50	0	0.0000
11	2	0.0041	31	3	0.0062	51	3	0.0062
12	5	0.0103	32	5	0.0103	52	13	0.0268
13	3	0.0062	33	4	0.0082	53	0	0.0000
14	2	0.0041	34	22	0.0454	54	4	0.0082
15	5	0.0103	35	9	0.0186	55	3	0.0062
16	8	0.0165	36	4	0.0082	56	9	0.0186
17	4	0.0082	37	16	0.0330	57	8	0.0165
18	0	0.0000	38	8	0.0165	58	5	0.0103
19	6	0.0124	39	10	0.0206	59	11	0.0227
20	4	0.0082	40	11	0.0227			

**Table 4 entropy-22-00699-t004:** The weights of each node.

Node	$\lambda_{i}$	Node	$\lambda_{i}$	Node	$\lambda_{i}$	Node	$\lambda_{i}$	Node	$\lambda_{i}$
1	0.0187	13	0.0062	25	0.0187	37	0.0333	49	0.0146
2	0.0146	14	0.0042	26	0.0146	38	0.0166	50	0.0000
3	0.0541	15	0.0104	27	0.0104	39	0.0208	51	0.0062
4	0.0437	16	0.0166	28	0.0125	40	0.0229	52	0.0270
5	0.0062	17	0.0083	29	0.0374	41	0.0520	53	0.0000
6	0.0042	18	0.0000	30	0.0873	42	0.0229	54	0.0083
7	0.0083	19	0.0125	31	0.0062	43	0.0042	55	0.0062
8	0.0062	20	0.0083	32	0.0104	44	0.0146	56	0.0187
9	0.0187	21	0.0146	33	0.0083	45	0.0125	57	0.0166
10	0.0146	22	0.0146	34	0.0437	46	0.0208	58	0.0104
11	0.0042	23	0.0249	35	0.0187	47	0.0104	59	0.0229
12	0.0104	24	0.0166	36	0.0083	48	0.0208

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
