# Peer review of "Risk Evaluation for a Manufacturing Process Based on a Directed Weighted Network"

_entropy, 2020, doi:10.3390/e22060699_

Round 1

Reviewer 1 Report

In the title of the paper, there is the term “risk”, while generally entropy characteristic is considered in the text of the paper (the latter one is encountered in the text 43 times against 23 times of the term risk). The correlation of these values is mentioned only in the middle of the paper. I think the title should be either corrected or the correlation should be considered in the first part of the paper. Besides, it should be clarified what kinds of risks are considered. For instance, if the risk is considered as an evaluation of actions’ destructive effects, then the review should be expanded respectively with references to the following methods of defining risks as sensitivity analysis, the analysis of break-even conditions, analogy method, probability theory-based methods (such as Monte-Carlo method), the method of adjusting initial data, etc. 

In section 3.1. the author distinguishes project phases. However, in the calculation example, there is an evaluation only for one phase. It is recommended to show how the general evaluation value is changed by the consideration of several phases. 

In the discussion section, it would be better to clarify how the knowledge gained by the author influences the efficiency of production performance and/or how to use the information which is given in the practical part. 

The paper would be much more significant if the author made highlights on what he suggests against the methods which he uses. The difference can be not only in the methods but also in the application. If that is the case, then it should be noted in the discussion section.

The reference 38 is better to be given as a resource link. 

Author Response

Dear Reviewer,

Thanks very much for taking your time to review this manuscript. I really appreciate all your comments and suggestions! Please find my itemized responses in below and my revisions/corrections in the re-submitted files.

Thanks again!

Comments to the author:

  1. In the title of the paper, there is the term “risk”, while generally entropy characteristic is considered in the text of the paper (the latter one is encountered in the text 43 times against 23 times of the term risk). The correlation of these values is mentioned only in the middle of the paper. I think the title should be either corrected or the correlation should be considered in the first part of the paper.

R1: Thank you for your suggestion. We added a description of the correlation between risk and information entropy. (Line 63-75.)

Moreover, the entropy H(I) of a single discrete random variable I is a measure of its average uncertainty. For the set of quality characteristics, the entropy of each quality characteristic represents the uncertainty of whether it can complete the requirements of the manufacturing process. That is, the centrality of each node indicates the uncertainty of whether the quality characteristics can meet the requirements of the manufacturing process in a certain time series. In addition, the quality of the manufacturing process actually refers to the degree to which a set of quality characteristics meet its production needs. Therefore, the centrality of each node is actually an indicator of the uncertainty of the quality characteristics in the manufacturing process, that is, a measure of quality loss. The greater the loss of quality in the manufacturing process, the greater the uncertainty of the manufacturing process. And the concept of risk was treated as synonym unreliability. Furthermore, the risk of the manufacturing process is characterized through the uncertainty, and the manufacturing process is characterized by two or more related quality characteristics. Therefore, the entropy value of the directed weighted network is determined according to the weighted average of the centrality of the nodes, and this value is defined as the risk of the manufacturing process.

  1. Besides, it should be clarified what kinds of risks are considered. For instance, if the risk is considered as an evaluation of actions’ destructive effects, then the review should be expanded respectively with references to the following methods of defining risks as sensitivity analysis, the analysis of break-even conditions, analogy method, probability theory-based methods (such as Monte-Carlo method), the method of adjusting initial data, etc.

R2: Thank you for your suggestion. We added a description of the kind of risk and the limitations of this paper.(Line 84-88)

Due to the random fluctuation of process factors, such as the degradation of machine, the aging of material, etc., which makes the quality characteristics are uncertain. Hence, these fluctuating process factors are considered as risk factors for the manufacturing process, but the focus of this paper is not on the mechanism by which these process factors affect the risk of the manufacturing process.

  1. In section 3.1. the author distinguishes project phases. However, in the calculation example, there is an evaluation only for one phase. It is recommended to show how the general evaluation value is changed by the consideration of several phases.

R3: Thank you for your suggestion. In the case of this paper, due to the limitation between the algorithm and the sample size, we explained the algorithm for a single stage. The improvement of the algorithm is the focus of our next research, and we also give instructions in our future work.

  1. In the discussion section, it would be better to clarify how the knowledge gained by the author influences the efficiency of production performance and/or how to use the information which is given in the practical part.

R4: Thank you for your suggestion. We further analyzed the calculated results and proposed risk control measures. In this paper, the focus is on introducing a new method of risk evaluation. Therefore, the identification of risk factors and the study of risk control strategies are the focus of our further research in the future. (Line 358-370)

According to equation (22), the risk of the manufacturing process is 0.6377. First of all, we should determine the acceptable threshold of risk according to the actual situation. If the risk is acceptable, no risk control measures are required for the manufacturing process. Otherwise, we should take risk control measures for the manufacturing process. Second, nodes 3, 4, 29, 34, 41, and 52 are identified as key nodes, and controlling the risks of these six nodes can effectively reduce the risk of the entire manufacturing process. According to the subnets of these six nodes, other quality characteristics that are causally related to these six quality characteristics can be quickly determined. We can change the processing technology, etc., so that the causal chain of quality characteristics is interrupted, that is, the propagation path of the risk is cut off. For example, node 3 affects node 29, and node 29 has an effect on other nodes. As long as the impact on node 3 on node 29 is cut off, the risk propagation path from node 3 to node 29 is interrupted, and the risk of node 3 is greatly reduced . Similar risk control measures are continuously taken for other quality characteristic causal chains until the risk of the manufacturing process reaches an acceptable level.

  1. The paper would be much more significant if the author made highlights on what he suggests against the methods which he uses. The difference can be not only in the methods but also in the application. If that is the case, then it should be noted in the discussion section.

R5: Thank you for your suggestion. We added the application scenario of the method in the discussion section. (Line 378-383)

Moreover, identifying how risk factors affect the quality of the manufacturing process is also a topic worth studying in future work. That is, what is the mechanism of the quality characteristic deviation caused by the process factors, which will have a great impact on the quality control. Furthermore, the PMIME method in this paper can be used for causality analysis between process factors and quality characteristics, which will make the research in this paper more significant.

  1. The reference 38 is better to be given as a resource link.

R6: Thank you for your suggestion. We have changed the citation format of reference 38.

Dua, D.; Graff, C. UCI Machine Learning Repository: SECOM Data Set Available online: http://archive.ics.uci.edu/ml/datasets/SECOM (accessed on Jun 9, 2020).

Reviewer 2 Report

Dear Authors, first of all let me express my compliments for the interesting subject of your research. I do not have serious negative considerations about your paper in its structure, results showed and analysis about them. 

I think the paper is interesting and it could be considered for publication. However, this paper can be further improved in the following points.

- the English level should be improved.
- the "Introduction" should be shorten or may be divided in two section (introduction and literature review);
- "future works" should be better explained

Please improve english and references, e.g.:
- Total Efficient Risk Priority Number (TERPN): a new method for risk assessment G. Di Bona, A. Forcina, A Silvestri, A.Petrillo
Journal of Risk Research Volume 21, Issue 11, 2 November 2018, Pages 1384-1408

Author Response

Dear Reviewer,

Thanks very much for taking your time to review this manuscript. I really appreciate all your comments and suggestions! Please find my itemized responses in below and my revisions/corrections in the re-submitted files.

Thanks again!

Comments to the author:

1.the English level should be improved.

R1: Thanks for your comments, and language errors or spelling errors have been corrected, e.g.(Line 36-39)

Quality function deployment (QFD) is a common theory and method for product design management driven by customer requirements [3]. The QFD is a process and set of tools used to effectively define customer requirements and convert them into quality characteristics.

  1. the "Introduction" should be shorten or may be divided in two section (introduction and literature review);

R2: Thanks for your comments. And the "Introduction" has been divided into two sections(introduction and literature review), which is section 1 and section 2.

  1. "future works" should be better explained

R3: We have added description of future works. (Line 378-383)

Moreover, identifying how risk factors affect the quality of the manufacturing process is also a topic worth studying in future work. That is, what is the mechanism of the quality characteristic deviation caused by the process factors, which will have a great impact on the quality control. Furthermore, the PMIME method in this paper can be used for causality analysis between process factors and quality characteristics, which will make the research in this paper more significant.

  1. Please improve english and references, e.g.:

- Total Efficient Risk Priority Number (TERPN): a new method for risk assessment G. Di Bona, A. Forcina, A Silvestri, A.Petrillo, Journal of Risk Research Volume 21, Issue 11, 2 November 2018, Pages 1384-1408

R4: We have increased the citation of the paper.(Cite 21)

Di Bona et al. [21] proposed a total efficient risk priority number method that integrated the failure mode, effects, and criticality analysis with other important factors in risk assessment.

Round 2

Reviewer 1 Report

I wish good luck to the authors in their future work. Please, in the future choose a more serious way to correct.

Author Response

Dear Reviewer,

Thanks very much for taking your time to review this manuscript. I really appreciate all your comments and suggestions! Please find my itemized responses in below and my revisions/corrections in the re-submitted files.

Thanks again!

Comments to the author:

  1. In the title of the paper, there is the term “risk”, while generally entropy characteristic is considered in the text of the paper (the latter one is encountered in the text 43 times against 23 times of the term risk). The correlation of these values is mentioned only in the middle of the paper. I think the title should be either corrected or the correlation should be considered in the first part of the paper. Besides, it should be clarified what kinds of risks are considered. For instance, if the risk is considered as an evaluation of actions’ destructive effects, then the review should be expanded respectively with references to the following methods of defining risks as sensitivity analysis, the analysis of break-even conditions, analogy method, probability theory-based methods (such as Monte-Carlo method), the method of adjusting initial data, etc.

    R1: Thank you for your suggestion. We added a description of the correlation between risk and information entropy.  (Line 63-77.)

    Moreover, the entropy H(I) of a single discrete random variable I is a measure of its average uncertainty. For the set of quality characteristics, the entropy of each quality characteristic represents the uncertainty of whether it can complete the requirements of the manufacturing process. That is, the centrality of each node indicates the uncertainty of whether the quality characteristics can meet the requirements of the manufacturing process in a certain time series. In addition, the quality of the manufacturing process actually refers to the degree to which a set of quality characteristics meet its production needs. Therefore, the risk of the manufacturing process is defined as the quality loss caused by the quality characteristics not meeting the production requirements. In this paper, the risk is evaluated by quantifying the uncertainty of the manufacturing process. When the uncertainty of the manufacturing process is greater, the more defects in the manufacturing process, the greater the quality loss. And the quality loss is invisible, which means that the reliability of the products produced by the manufacturing process is relatively low. Moreover, the quality loss will spread over the connection between quality characteristics, so the connection between quality characteristics is actually the risk propagation path of the manufacturing process.

  2. In section 3.1. the author distinguishes project phases. However, in the calculation example, there is an evaluation only for one phase. It is recommended to show how the general evaluation value is changed by the consideration of several phases.

    R2: Thank you for your suggestion. In the case of this paper, due to the limitation between the algorithm and the sample size, we explained the algorithm for a single stage. The improvement of the algorithm is the focus of our next research, and we also give instructions in our future work.

  3. In the discussion section, it would be better to clarify how the knowledge gained by the author influences the efficiency of production performance and/or how to use the information which is given in the practical part.

    R3: Thank you for your suggestion. We further analyzed the calculated results and proposed risk control measures. Due to the large amount of content, the specific content can be seen in the modified paper(Line 358-385)

  4. The paper would be much more significant if the author made highlights on what he suggests against the methods which he uses. The difference can be not only in the methods but also in the application. If that is the case, then it should be noted in the discussion section.

    R4: Thank you for your suggestion. We added the application scenario of the method in the discussion section. (Line 393-398)

    Moreover, identifying how process factors affect the quality of the manufacturing process is also a topic worth studying in future work. That is, what is the mechanism of the quality characteristic deviation caused by the process factors, which will have a great impact on the quality control. Furthermore, the PMIME method in this paper can be used for causality analysis between process factors and quality characteristics, which will make the research in this paper more significant.
